# Isolation, Propagation and Genotyping of Human Rotaviruses Circulating among Children with Gastroenteritis in Two Egyptian University Hospitals

**DOI:** 10.3390/biology11101413

**Published:** 2022-09-28

**Authors:** Mona H. El-Gayar, Sarra E. Saleh, Aly F. Mohamed, Mohammad M. Aboulwafa, Nadia A. Hassouna, Abdou Kamal Allayeh

**Affiliations:** 1Department of Microbiology and Immunology, Faculty of Pharmacy, Ain Shams University, Organization of African Unity St., Cairo 11566, Egypt; 2The International Center for Training & Advanced Researches (ICTAR–Egypt1), Cairo 12622, Egypt; 3Faculty of Pharmacy, King Salman International University, Ras-Sudr 46612, Egypt; 4Virology Laboratory, Water Pollution Research Department, Environment and Climate Change Research Institute, National Research Center, Giza 12622, Egypt

**Keywords:** rotavirus, isolation, propagation, characterization, Egypt

## Abstract

**Simple Summary:**

Group A rotaviruses are the most common cause of acute gastroenteritis affecting Egyptian children under the age of five, with symptoms ranging from asymptomatic infection to severe dehydration or death. In the present work, diarrheal samples from Egyptian children admitted to gastrointestinal pediatric wards of two main governmental hospitals were collected and molecularly analyzed for Group A rotavirus. Our findings revealed that rotaviruses accounted for more than one-sixth of all cases under study, peaking in the winter. G1P[8] was the most prevalent rotavirus genotype in this study. The two cell lines used in our work coherently isolated and propagated rotavirus strains. Continuous rotavirus detection and genome sequencing of the successfully isolated strains will be recommended in the future in order to support the control of such viruses, and tackle the problem in Egypt.

**Abstract:**

The most prevalent cause of infectious neonatal diarrhea is Group A rotavirus (RVA). Unfortunately, there is a dearth of data on the incidence of rotavirus-associated infections among Egyptian children. The present study aimed to isolate, propagate, and genotype human rotaviruses circulating among Egyptian children with acute gastroenteritis admitted to two main university pediatric hospitals, Abo El-Reesh and El-Demerdash, over two consecutive winters, 2018–2020. Diarrheal samples (*n* = 230) were screened for Group A rotavirus RNA using RT-PCR assay. In positive samples (*n* = 34), multiplex semi-nested PCR was utilized to determine G and P genotypes. Thirty-four (14.8%) of the collected samples tested positive. The genotype distribution revealed that G1P[8] was the predominant rotavirus genotype throughout the current study. All rotavirus-positive fecal samples were passaged twice on human colorectal adenocarcinoma cell line (Caco-2) and rhesus monkey kidney epithelial cell line (MA104). Both cell lines could successfully isolate 14.7% (*n* = 5 out of 34) of the identified strains; however, Caco-2 cell line was shown to be more efficient than MA104 in promoting the propagation of human rotaviruses identified in Egyptian children’s feces.

## 1. Introduction

Human rotavirus (RV) infections are still the primary cause of acute gastroenteritis in infants and children under the age of five in both developed and developing countries, with symptoms ranging from asymptomatic infection to severe dehydration or death [1]. According to epidemiological studies, this RNA virus is the top leading cause of non-bacterial gastroenteritis, accounting for more than 50% of all occurrences of intestinal viral infection in children [2,3,4]. Globally, rotaviruses cause about 114 million episodes of self-limiting acute gastroenteritis, 24 million clinic visits, 2.4 million hospitalizations, and more than half million deaths in infants and young children every year [5,6]. RVs are nonenveloped and double-stranded RNA viruses that belong to the Reoviridae family [7]. The virion consists of three protein shells surrounding 11 segments of dsRNA encoding six structural proteins (VP1-VP4, VP6, and VP7) as well as six nonstructural proteins (NSP1-NSP6) [8]. Based on the sequences of VP7 (G-type antigen) and VP4 (P-type antigen), RVs could be classified into different genotypes. Until now, 42 G-types and 58 P-types have been described worldwide [9]. RVs are classified based on a group antigen in VP6 capsid protein into nine groups, designated as A–D and F–J [10]. Group A rotaviruses (RVAs) are the most prevalent, and they are to blame for children’s hospitalizations worldwide [5,10,11]. In Egypt, they are responsible for 40–60% of fall and winter cases [12] and research has shown that clean water supplies and improvements in sanitation have minute effects in preventing RVA-induced diarrhea [6,13]. Two live, attenuated RV vaccines are commercially available and are being introduced into countries across the world. Rotarix^®^ (GSK Biologicals, Rixensart, Belgium, prequalified in 2009) is an attenuated G1P[8] human RV strain that has demonstrated high vaccination efficacy in Europe and Africa. RotaTeq^®^ (Merck & Co. Whitehouse, New Jersey, USA, prequalified in 2008) is a bovine human RV reassortant vaccine that has been shown to be effective in the United States, Europe, and Africa [14,15]. Rotavac (Bharat Biotec, Hyderabad, India, prequalified in 2018) and Rotasiil (Serum Institute of India, Pune, India, prequalified in 2018) are two other attenuated vaccines that are only in use in India (both vaccines) and Palestine (Rotavac only) [16,17]. Circulating RVs are antigenically varied, both geographically and temporally, and there are worries that the vaccination may not protect against all strains that cause viral infection [18]. Africa and Asia, without a doubt, have a greater strain variety and variability than Europe or the United States [19]. As RV vaccinations are being launched internationally, it is critical to examine the diversity of strains circulating in countries such as Egypt. Rotavirus detection generates valuable data on the circulating RV strains. These data are vital to improve vaccine development, tracking emergent types and helping to assess vaccine effectiveness, and changes in strains diversity after vaccines have been introduced [20]. Therefore, the objectives of this study were to identify the genotypes of the circulating RV strains in a cohort of Egyptian children ranging from birth to five years of age admitted to two main university pediatric hospitals, as well as to compare isolation efficiency of human rotaviruses from Egyptian infants using human colorectal adenocarcinoma (Caco-2) and rhesus monkey kidney epithelial (MA104) cell lines.

## 2. Materials and Methods

### 2.1. Specimen Collection and Study Design

This is a cross-sectional study in which samples were collected from many different individuals at a specific time and variables were observed without influencing them. Two university pediatric hospitals in Egypt, Abo El-Reesh and El Demerdash, each with roughly 3500 beds, were selected as specimen collection sites. These are two primary public hospitals in the Great Cairo area, serving millions of children from Egypt’s urban, semi-urban, and rural areas, as well as serving as referral hospitals for patients from all over the country. Furthermore, they are public hospitals with gastrointestinal pediatric wards that serve a broad community. The fecal samples were obtained from children under the age of five who were hospitalized or visited the emergency pediatric wards of the aforementioned hospitals over two consecutive winters (December 2018 to April 2020) (Figure 1). Samples were gathered in clean collecting containers and transferred to Ain Shams University’s microbiology and immunology laboratories under cold circumstances.

### 2.2. Ethical Approval and Consent

After receiving written consents from the participants’ guardians, the sampling was carried out. In compliance with the Declaration of Helsinki principles, the ethics committee of the Faculty of Pharmacy, Ain Shams University in Cairo, Egypt approved this study (ENREC-ASU-2019-83).

### 2.3. Demographic Data Collection and Specimens Processing

Clinical data such as the children’s age, gender, place of accommodation, date of hospitalization, date of fecal collection, and frequency of diarrhea (number of episodes per day) were all collected. For each specimen, 1 g of feces was weighed, diluted 1:10 in phosphate buffered saline (PBS), pH 7.2, and vortexed for 1 min. At room temperature, samples were cleaned by centrifugation at 7000 rpm for 10 min. The recovered supernatant was utilized to extract viral RNA. For isolation, the acquired supernatants were filtered through 0.22 µm syringe filter (Catalog number: SCA020025K-S), (CHM LAB, Terrassa, Barcelona, Spain) and kept at −80 °C until use [21].

### 2.4. Viral RNA Extraction

The RNA was extracted from membrane-filtered (0.22 µm) stool supernatant using the GeneJET RNA Purification kit (catalog number: K07321) from Thermo Scientific (Waltham, MA, USA) according to the manufacturer’s instructions. For subsequent investigation, the isolated dsRNA was kept at −20°C [22].

### 2.5. Molecular Detection of Rotaviruses Using VP6 Segment

The extracted dsRNA was utilized as a template for one-step RT-PCR, with reverse transcription and PCR amplification of the VP6 fragment using one-step MaximeTM RT-PCR PreMix (catalog number: 25131), (iNtRON Biotechnology, Seoul, Korea) according to the manufacturer’s instructions. To amplify the 824-bp VP6 fragment, a primer set (forward primer VP6-F-5′-GGCTTTWAAACGAAGTCTTC-′3 and reverse primer VP6-R-5′-GGTCACATCCTCTCACT-′3) was utilized as described by Matthijnssens et al. [23]. The PCR settings are included in the Appendix A, and the PCR products were identified by agarose gel electrophoresis (1.5% agarose in Tris-acetic acid-EDTA buffer (0.089 M Tris, 0.089 M acetic acid, 0.002 M EDTA, pH 7.5), containing 0.5 μg/mL ethidium bromide). Under a UV transilluminator, amplicons stained with ethidium bromide were examined and the sizes of the DNA fragments were determined by comparison to a 100 bp DNA ladder (GeneRuler 100 bp DNA ladder, Thermo Fisher Scientific, Waltham, MA, USA).

### 2.6. Cell Lines and Positive Rotavirus Samples Isolation and Propagation

The rhesus monkey kidney epithelial cell line (MA104), (ATCC, Manassas, VA, USA) and human colorectal adenocarcinoma cell line (Caco-2), (ATCC, Manassas, VA, USA) were kindly provided by Dr. Abdou K. Allayeh, (Environmental Virology Laboratory, Water Pollution Research Department, National Research Centre (NRC), Egypt). The Caco-2 cell line was first isolated in 1977 from a 72-year-old Caucasian male colonic tumor [24]. Upon differentiation, those cells showed morphological and biochemical characteristics of small intestine enterocytes. Caco-2 is considered an excellent in vitro model for intestinal epithelium [25]. MA-I04 cells are an established cell line derived from fetal rhesus monkey kidney and the first line to isolate rotaviruses with high selectivity [26]. All PCR positive rotavirus samples were subjected to isolation and propagation using both cell lines. For viral activation, each sample was incubated at 37 °C for 30 min following gentle mixing with 1 µg/mL of TPCK-Treated Trypsin (catalog number: 20230), (Thermo Scientific, Waltham, MA, USA). Simultaneously, DMEM maintenance medium containing 2% fetal bovine serum (FBS) was utilized. To ensure that the monolayers were properly infected, the growth media was removed and the cell monolayers were washed three times with 5 mL of sterile pre-warmed PBS. This step is critical since FBS suppresses RV activation (proteolytic priming) [27]. After activation, each sample was inoculated into a confluent monolayer of MA104 and Caco-2 cells, gently mixed and incubated in a CO_2_ incubator at 37 °C for 30 min to allow viral adsorption. The flasks were then filled with 10 mL of pre-warmed DMEM medium. Finally, the infected flasks were checked microscopically on a daily basis for the existence of cytopathic effect (CPE); cells shrank rapidly, became dense in a process known as pyknosis, detached from the glass, and died [28]. For further propagation, the infected cells were freeze-thawed three times and inoculated onto fresh confluent monolayer cells for another passage after virus activation with treated trypsin as described above. Usually, RV CPE is observed after multiple passages, however, the absence of CPE after six passages, considering RV isolation negative [29,30]. Cell monolayers were photographed using an inverted microscope (Olympus Corporation) equipped with a digital camera. For virus preservation, the infected cells were freeze-thawed three times, then centrifuged, and the clarified virus supernatants were aliquoted and kept at −80 °C for long-term preservation [31].

### 2.7. Molecular Typing of Positive Samples Using VP7 and VP4 Segments

With slight modifications, the VP7 and VP4 genotyping was performed using a multiplex RT-PCR technique reported by Gouvea et al. [32] and Gentsch et al. [33]. To amplify the genes of interest, a reverse-transcription step was followed by a PCR step using specific primer pairs ((Beg9/End9) for VP7 and (Con2/ Con3) for VP4) as terminal primers. Afterwards, a semi-nested PCR was performed using specific primers to detect the viral genotype (see Appendix B and Appendix A). The amplicons were separated using 1.5% agarose gel electrophoresis, and the genotype was determined based on the product size [34]. The band sizes of the predicted G- and P-types, which were utilized to interpret human rotavirus G/P typing, are provided in in the Appendix A [32,33,34].

### 2.8. PCR Products Sequencing

The MEGA total fragment DNA purification kit (iNtRON Biotechnology, Seoul, Korea) was used to purify PCR products of a selected rotavirus strain, RVA144, as directed by the manufacturer. The nucleotide sequences were analyzed using an ABI Prism^®^ 310 Genetic Analyzer (Colors-Lab, Cairo, Egypt). An aliquot (10 µL) of each fragment was sequenced from both the forward and reverse directions with the same primer pairs listed in the PCR section (VP6-F/VP6-R for VP6, Beg9/End9 for VP7, Con2/Con3 for VP4), using an ABI Prism Big dye termination cycle sequencing kit (Applied Biosystems, Foster City, CA, USA). Using the BioEdit v7.2.5 software, the obtained ABI sequence files were assembled into final contigs [35]. The National Center for Biotechnology Information (National Institutes of Health, Bethesda, MD, USA) BLASTn (basic local alignment search tool) server was used to search the GenBank database for nucleotide sequence similarity (https://blast.ncbi.nlm.nih.gov/Blast.cgi, accessed on 15 December 2021). Molecular Evolutionary Genetics Analysis (MEGA) X software v.10.1.7 was used to calculate multiple sequence alignments [36]. The VP6, VP4, and VP7 nucleotide sequences were aligned to those of human rotaviruses in the GenBank database before being submitted to the NCBI and assigned to accession numbers.

### 2.9. Nucleotide Sequence Accession Numbers and Phylogenetic Tree Construction

The following are the GenBank accession numbers for the submitted sequences described in this study: OL963766, OL963767, and ON152314. The acquired VP6, VP4, and VP7 sequences were aligned and compared to the other rotaviruses, VP6, VP4, and VP7, sequences in the Genebank database (www.ncbi.nlm.nih.gov/genbank/, accessed on 21 December 2021). To investigate the genetic relatedness of the selected rotavirus strain (RVA144) in the present study to other rotavirus strains in the Genebank database, phylogenetic analyses for VP6, VP4, and VP7 genes were generated by the Maximum Likelihood (ML) approach using MEGA X software v.10.1.7 [36]. The tree was drawn to scale, with branch lengths measured in the number of substitutions per site. This analysis involved 100 nucleotide sequences. Phylogenetic distances were evaluated by the Tamura-Nei model using 1000 iterations for the bootstrap analyses implemented in the Molecular Evolutionary Genetics Analysis (MEGA) X software v.10.1.7 (www.megasoftware.net/, accessed on 23 March 2022) [36,37].

### 2.10. Statistical Analysis

For statistical analysis, the GraphPad Prism application version 5.0 was used. The Chi-square test was used to analyze the significance of rotavirus infection rates across gender and seasons as well as variations in viral infection frequencies among age groups. All tests were two-tailed, testing for the possibility of the relationship in both directions. If the *p**-**value* was *0.05* or lower, the *result* was labeled as *significant*.

## 3. Results

### 3.1. Virologic Detection of Rotaviruses in Clinical Samples

As shown in Table 1, a total of 230 fecal samples (128 from males and 102 from females) were obtained from children under the age of five who were hospitalized in the emergency pediatric wards of two major pediatric hospitals (Abo El-Reesh and El Demerdash) in Cairo, Egypt. Only 23 (18%) male and 11 (10.8%) female cases tested positive for rotavirus, with an age mean of 0.92. Most of the positive rotavirus cases were in children under the age of two years. A total of 25 of them (28.0%) were less than one year old (76% of them in the age group 6–12 months) compared to 64 negative RV cases. A total of 8 RV cases (6.0%) were between one and two years old, compared to 125 negative RV cases. Except for the summer season, rotavirus was discovered throughout the year; nevertheless, there was an evident seasonality of rotavirus diarrhea, with a larger prevalence in the winter (18.1%), followed by spring (15.1%), and autumn (7.3%). The present cross-sectional study included more children from metropolitan regions (Cairo and Giza) rather than the rural areas in either the RV (34 cases) or the non-RV (196 cases) groups, with no significant difference between the two groups.

### 3.2. Isolation and Propagation of Detected Rotavirus Using Different Cell Lines

Caco-2 and MA104 cells were infected with a subset of the identical 34 rotavirus-positive fecal samples and passaged twice. The CPE began to appear as the subculture continued, and the time needed for a lesion to appear gradually decreased as the number of passages increased, indicating a gradual increase in virus infectivity. In Caco-2 cells, 5 out of 34 (14.7%) RV strains were isolated on the first passage, and all 5 strains passed the second and third passage well. On the other hand, MA104 cells did not support the growth of the virus on the first passage. However, 3 out of the 34 (8.82%) strains identified on MA104 cells grew on the second passage, while only 2 grew on the third one. Finally, 5 of the 34 strains developed CPE in Caco-2 cells on the first and second passages (Table 2). It should be noted that the strains (*n* = 5) propagated successfully on Caco-2 cells were the same as the strains propagated on MA104 cells. Caco-2 cultures supported rotavirus proliferation of a substantially higher number of isolates than MA104 cells in the first and second passages. It should be emphasized that the isolated RVA strains exhibited different degrees of CPE, where some strains showed complete (total) destruction of the mammalian cell monolayer while others produced partial (subtotal) destruction of the monolayer (Table 3, Figure 2). As shown in Table 3, RVA144 was the only strain that caused complete destruction for both cell lines mentioned above and exhibited the most severe form of CPE, where all cells in the monolayer became rounded and detached from the glass within three days.

### 3.3. G/P Typing of the Isolated Rotavirus Strains

The G- and P-types could be attributed to 34 (14.7%) rotaviruses out of the 230 diarrheal episodes investigated. Among the typed strains, G1 was the only recognized G genotype (100%) and P[8] was the only detected P genotype (100%). Ultimately, rotavirus G1P[8] was found to be the single genotype detected in the clinical samples of the present study.

### 3.4. Phylogenetic Tree Analysis

When the VP6, VP4, and VP7 segments of the isolated strain RVA144 were analyzed, they were closely related to those of human RVA strains and particularly had human Wa-like constellations with nucleotide similarity up to 99.78, 99.66, and 99.90%, respectively. They were clustered with all human RVA strains in the phylogenetic analysis. As shown in Figure 3, the RVA144 VP6 segment was closely related to human RVA Wa strain in the USA and South Africa. Regarding the VP4 and VP7 segments, they were also closely related to human RVA Wa strain in the USA as well as those identified in USA labs.

## 4. Discussion

The current study is one of a handful of research projects on rotavirus disease in children under the age of five that has been conducted in Egypt [12,21,38,39,40]. This study reports the distribution of rotavirus isolated from fecal samples in a repeated cross-sectional study of 230 Egyptian children, ranging in age from birth to five years, between 2018 and 2020 in two major university hospitals in Egypt. These are public hospitals that serve a broad community; therefore, the current study’s findings will most certainly reflect a sizable section of the country’s population. Our findings revealed that rotaviruses are important aetiologic agents of severe diarrhea, accounting for over one sixth of all cases with acute gastroenteritis, with an incidence of 14.8% (confidence interval 10.4–20.0). There was no significant difference in RV positivity between male and female genders. However, the odds of infection were 1.8 times higher among males compared to females. This observation was in accordance with a prior study in Egypt that reported higher RV detection rate in males compared to females [12]. There were 34 children in the study who had at least one episode of rotavirus diarrhea, with younger children (<1 year) suffering from the disease at a higher incidence (28%) than the older ones (18.5%), and the difference was statistically highly significant. These results go beyond previous Egyptian reports conducted between 2004 and 2007, 2011 and 2012, and 2015 and 2016, showing that the majority of rotavirus cases occurred during the first year of life [21,38,39]. On the other hand, this greater rotavirus prevalence in Egyptian children under the age of one year is higher than that reported in other developing countries, such as India and Mozambique [41,42]. This disparity in incidence rates between Egypt and other developing nations may be attributable to Egypt’s lack of a national rotavirus vaccination program. It was also noticed that rotavirus showed higher existence in the age group 6–12 months compared to younger ones. This may be attributed to the high activity of children during these months, the beginning of walking age, as well as the dentation, that may lower the immunity at this period, increasing their risk of infection, whereas children younger than that age have a high level of maternal immunity [43]. In the current study, rotavirus infections peaked in winter (18.1%), followed by spring (15.1%), and autumn (7.3%), which is similar to the winter peak reported in a sectional study conducted in 2012 [39], but differs from the seasonal peak reported in another one conducted in 2018 [21]. This disparity might be attributed to climate change or other contributing variables such as urban residency. In Egypt, the distribution of rotavirus genotypes has shifted over the last few years. In 2015–2016, G1P[8] in Cairo was 29.7% [21]. Another study conducted between 2015 and 2017 discovered that G1 was detected in 26% of cases, with the P genotype P[8] 22.8% [12]. Surprisingly, in our study, the typeable genotypes were G1P[8] 100% of the time, which is a significant difference when compared to previous publications in Egypt. The fact that this study only included a subset of inpatients and outpatients admitted to two hospitals for diarrhea treatment is one of its limitations. Despite the fact that the aforementioned hospitals serve as referral hospitals for patients from all over the country, rural samples still did not exceed 19.7% of total samples collected. The present study was restricted to two hospitals in the urban regions and did not extend to rural areas where higher genotype diversity may be detected due to interspecies rotavirus transmission, which may explain our findings [38,44,45]. On the other hand, the predominance of only one genotype does not necessitate the absence of high mutation rates in the virus genome. Tracking G/P genes only in the present study did not provide a comprehensive picture on the entire virus genome because the dual typing system (i.e., G/P genes) is restricted to outer capsid encoding genome segments. Therefore, a whole genome-based genotyping would be recommended as it is based on the assignment of genotypes to all the 11 gene segments (i.e., G/P and non-G/P genes) [46]. Regarding rotavirus isolation, both Caco-2 and MA104 cell lines successfully support the isolation and propagation of the virus. Our findings were in accordance with a study conducted in 2020 that confirmed that both Caco-2 and MA104 cells were used for the isolation and propagation of RVA strains [47]. It is worth noting that the complete CPE could not be observed on Caco-2 and MA104 cells from the first passage. In accordance with our findings, a study performed in 2016 by Ennima at al. reported that RV CPE was not observed after single passage and there was an absence of CPE after six passages, considering RV isolation negative [30]. Another study reported that RV CPE began to appear on MA104 cells as subculture continued into the fourth passage [28]. Our findings also concluded that Caco-2 cultures supported rotavirus proliferation of a substantially higher number of isolates than MA104 cells in the first and second passages. This observation is in accordance with that deduced in 2016, where most Caco-2 passaged rotaviruses were grown in a higher proportion than MA104 cells [48]. In the present study, we successfully adapted five rotavirus isolates to grow and induce cell death in colonic (Caco-2) tumor cells, which represent one of the most epidemiologically important human cancers. This remarkable property of the isolated rotaviruses makes them promising candidates for use as oncolytic agents, in the future [49]. Phylogenetic analysis revealed that the VP6, VP4, and VP7 segments of RVA144 isolated from Egyptian clinical specimens belong to G1P[8]. Our dataset indicated a high similarity to those of human RVA strains and particularly human Wa-like strains in the USA. These findings were in accordance with another studies in Egypt that reported the similarity between the VP6 and VP4 genes of RVA isolated from clinical and environmental samples and the Wa reference strain in the USA [12,40]. The low variations in VP6 sequences contained in the Egyptian G1P8 genotype may suggest that VP6 recombinant subunit vaccines would be useful candidates for rotavirus vaccination among Egyptian children. Unlike other studies, our data did not support any relatedness to other rotavirus species [50]. The current findings emphasize the importance of the continuous surveillance of the circulating rotavirus among Egyptian children in urban and rural regions, which is important for monitoring virus emergence and assisting in predicting the protection level afforded by rotavirus vaccines. Our future perspectives include whole genome sequencing of all successfully rescued rotavirus strains to track all mutations on such genome, as well as attempting to use these strains as candidates for preparation of a protective vaccine against rotavirus infections to overcome the issue in Egypt.

## 5. Conclusions

Rotavirus diarrhea in children is still considered a public health issue in Egypt. According to our findings, RVs accounted for more than one-sixth of all pediatric instances of acute gastroenteritis. This work identified a single rotavirus genotype, G1P[8], and proved that utilizing Caco-2 and MA104 cell lines is useful for viral isolation and multiplication. Continuous surveillance of RV in the Egyptian population, whole genome sequencing of the rescued RV strains, as well as preparation of a protective vaccine to defend against this virus will be recommended in the future.

## Figures and Tables

**Figure 1 biology-11-01413-f001:**
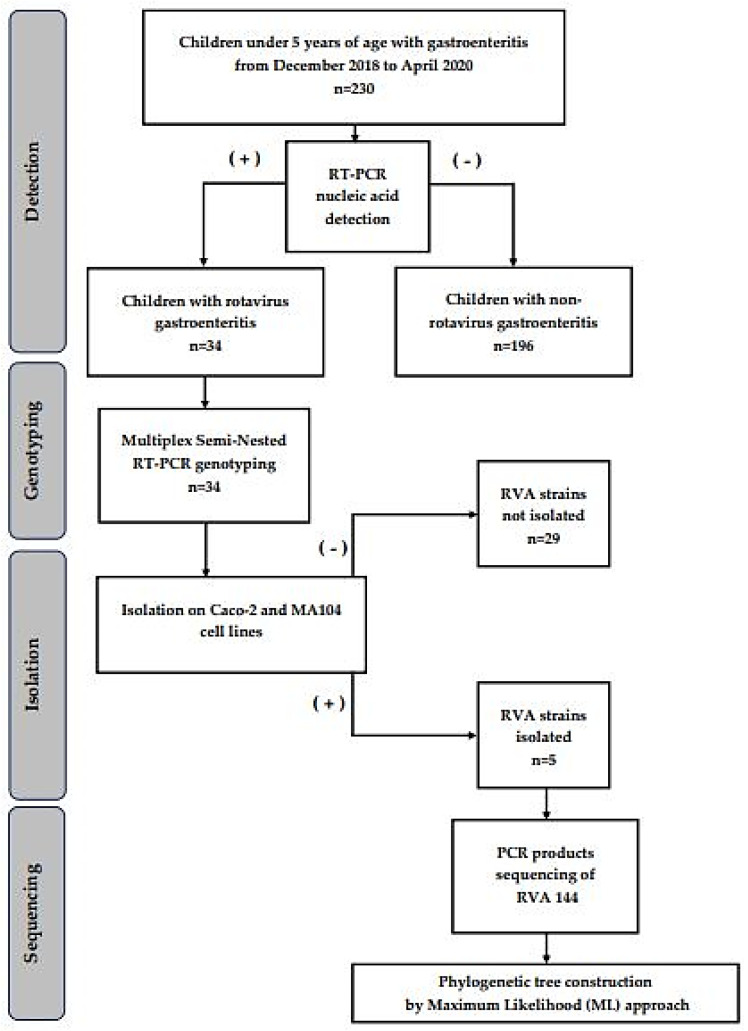
Flow chart summarizing the cross-sectional study of rotavirus cases among children with gastroenteritis in two Egyptian University Hospitals.

**Figure 2 biology-11-01413-f002:**
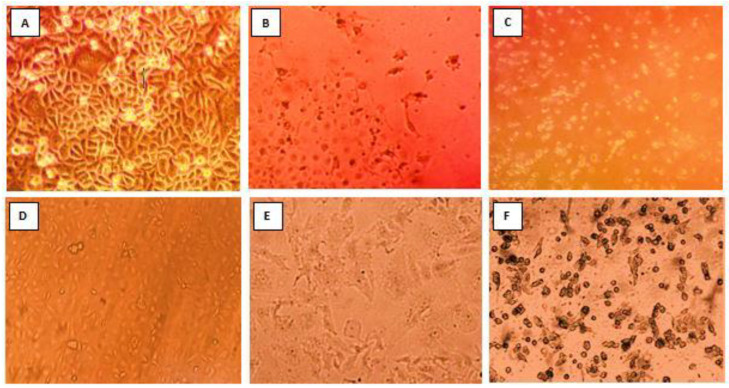
Development of cytopathic effect on Caco-2 and MA104 cell lines during isolation of rotavirus: (**A**) control Caco-2 cell line after 3 days of propagation; (**B**) infected Caco-2 by positive sample of rotavirus showing partial CPE; (**C**) infected Caco-2 cells by positive sample of rotavirus showing complete CPE; (**D**) control MA104 cells after 3 days of propagation; (**E**) infected MA104 by positive sample of rotavirus showing partial CPE; (**F**) infected MA104 by positive sample of rotavirus showing complete CPE.

**Figure 3 biology-11-01413-f003:**
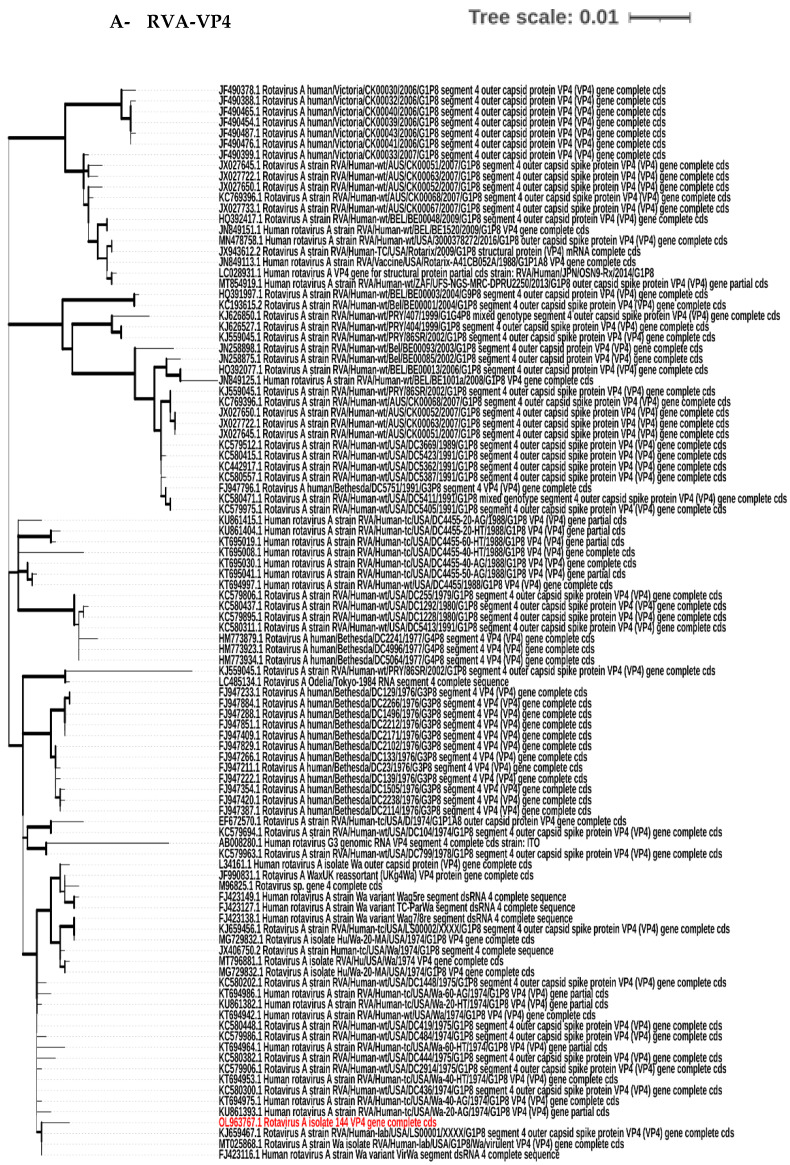
Phylogenetic trees of VP4 (**A**), VP6 (**B**) and VP7 (**C**) nucleotides’ sequences of RVA144 strain (labelled by the red color) detected among Egyptian children rotavirus cases showing their genetic relationship with the other human *RVA*s. This analysis involved 100 nucleotide sequences. The tree was generated by the Maximum Likelihood (ML) method with bootstrap (1000 iterations) analyses using the MEGA X software v.10.1.7 [36].

**Table 1 biology-11-01413-t001:** Distribution of positive rotavirus (RV) samples in relation to gender, age, season, and residency.

Variable		RV Positive *n* = 34 N (%)	RV Negative *n* = 196 N (%)	*p*-Value
Gender	Male	23 (18.0)	105 (82.0)	0.130
	Female	11 (10.8)	91 (89.2)	
Age	≤1 year	25 (28.0)	64 (72.0)	<0.001 **
	>1 year–2 years	8 (6.0)	125 (94.0)	
	>2 years	1 (12.5)	7 (87.5)	
Season	Autumn	3 (7.3)	38 (92.7)	0.100
	Spring	8 (15.1)	45 (84.9)	
	Summer	0 (0)	9 (100)	
	Winter	23 (18.1)	104 (81.9)	
Residence	Cairo	15 (15.6)	81 (84.4)	0.300
	Giza	17 (16.2)	88 (83.8)	
	Others	2 (6.9)	27(93.1)	

** *p* < 0.001 highly significant.

**Table 2 biology-11-01413-t002:** Propagation of positive rotavirus samples * using MA104 and Caco-2 cell lines.

Cell Line	Passage Number	No. of RVA Strain(s) Isolated	RVA Strain (s)
MA104	First passage	0	None
	Second passage	3	RVA144, 113, 206
	Third passage	2	RVA 111, 163
Caco-2	First passage	5	RVA144, 113, 163, 111, 206
	Second passage	5	RVA144, 113, 163, 111, 206
	Third passage	5	RVA144, 113, 163, 111, 206

* The number of positive rotavirus samples detected by PCR was 34.

**Table 3 biology-11-01413-t003:** Cytopathic effect of the isolated RVA strains.

RVA Strain	MA104	Caco-2
144 *	Complete	Complete
113	Partial	Complete
206	Complete	Partial
111	Partial	Partial
163	Partial	Partial

* RVA144 was the only strain that caused complete destruction for both cell lines.

## Data Availability

All the data supporting the findings are included in the manuscript and Appendix A.

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
