# Peer review of "Isolation, Propagation and Genotyping of Human Rotaviruses Circulating among Children with Gastroenteritis in Two Egyptian University Hospitals"

_biology, 2022, doi:10.3390/biology11101413_

Round 1

Reviewer 1 Report (Previous Reviewer 1)

P2, L58: Number of accepted G- and P-genotypes for RVA can be found on official web pages of Rotavirus Classification Working Group - RCWG (https://rega.kuleuven.be/cev/viralmetagenomics/virus-classification/rcwg). Currently (as of on July 6th, 2022) there were 42 assigned G-genotypes and 58 P-genotypes. Please, check the web page, correct the numbers of genotypes and change the appropriate citation.

P2, L60: Currently, there are nine RV species, Rotavirus A-D,F-J. Rotavirus E was created in 1999 based on genome profiling and comparative terminal fingerprinting (Pedley et al., 1986). However, due to the lack of any subsequent virus isolates or sequence data, it was removed as a species by the ICTV in 2019 (https://ictv.global/report/chapter/sedoreoviridae/sedoreoviridae/rotavirus). Please, correct the information and add an appropriate citation.

P3, L118: The citation No. 22 (Rizk and Allayeh, 2020) is not appropriate. Please, remove the citation or use it in appropriate context.

Figure 2: The phylogenetic trees are still of very poor quality. It is not possible to publish trees where the names of strains are not discernible. I strongly recommend the authors to save the trees as a file with higher resolution (e.g. TIF file).

Author Response

Reviewer 2 Report (New Reviewer)

In this manuscript, the authors collected 230 clinic samples from two hospitals in Egypt, and analyzed them. They found 14.8% of the samples are rotaviruses positive and identified the genotypes of rotaviruses from positive samples, and used the samples to infect Caco-a and MA104 cell lines. The whole story of the research is very simple. The authors did not solve any scientific questions or put up any discoveries in their research.

Author Response

Reviewer 3 Report (New Reviewer)

El-Gayar et al performed a study regarding the prevalence of rotavirus in children under five years admitted to two Egyptian hospitals and their ability to damage a colorectal carcinoma cell line. They found a prevalence of 14.8% in their cohort containing 230 samples and they found that in 5 of 34 samples the virus was able to grow within both the cell line Caco-2 and MA104.

In my opinion, the manuscript is complicated to understand as two main topics are covered: The prevalence of Rotavirus and the growing ability of that virus in two cell lines. I think that it would be the best to either concentrate on one of those very important findings or to split all analyses to two manuscripts. When concentrating on the prevalence, it would be favourable to add data of the outcome (time of admission, complications, death) into the analysis. Was there a worse outcome in patients infected with Rotavirus?

The manuscript is written in adequate English and besides of questionable interpretation (see below) statistics seem to be adequately performed. Citations are adequate.

Minor comments:

-        Abstract: Line 34 “[…] whereas 196… are negative”. Please remove (redundant).

-        Abstract: Line 35 “Multiplex Semi-Nested…”: Belongs to the Methods, please move some sentences up.

-        Materials and Methods: I encourage the authors to add a sentence containing the study’s details: “This study is a cross-sectional study….” At the beginning of this section.

-        Lines 93-94: “As a result, …” This sentence may be moved to the Discussion section.

-        The Methods section is quite long, is there a possibility to move a part of the details to the Appendix?

-        Line 219: Was p-value two-sided?

-        Table 1: Please provide the same number of decimals for each p value.

-        Lines 240-247: This part may be moved to the Methods section.

-        Line 257: If “the difference was not statistically significant”, it means that there was no difference.

-        Lines 257-260: Descriptions of pyknosis may be defined in the Methods section.

-        Lines 273-277 may be moved to the Methods section / Appendix.

-        Figure 2 cannot be read, please provide adequate quality.

-        Lines 314-315: “There was no significant difference […] between male and female gender […]. However, the higher prevalence was noticed among males […]” The authors should stick to their statistical findings (no difference).

-        Do the authors have any data on the total number of admissions during their observation period?

-        Lines 386-395: The Conclusion is quite long.

Round 2

Reviewer 2 Report (New Reviewer)

I accept the current revised version of it.

Author Response

Reviewer 3 Report (New Reviewer)

The authors responded to my comments in the "Response to authors" but did not edit their manuscript adequately. For example, in the current manuscript th two-sided p value is not explained. Furthermore, the authors did not put a sentence explaining the methodology of the study to the beginning of the "Methods" section as suggested. They also did not move any text from the Methods section to the appendix as suggested. Finally, I partially agree with the other reviewers regarding the questionable novelty of this paper's results.

Round 3

Reviewer 3 Report (New Reviewer)

I congratulate the authors and have no further comments.

This manuscript is a resubmission of an earlier submission. The following is a list of the peer review reports and author responses from that submission.

Round 1

Reviewer 1 Report

The manuscript by El-Gayar, Saleh, Mohamed et al. brings results of detection and propagation of human rotavirus A in children under 5 years in Egypt. During two years there were 34 RVA-positive samples all of G1P[8] genotype. The whole sequences of VP7, VP4 and VP6 genomic segments of one RVA isolate have been deposited in the GenBank. Five RVA-positive samples were successfully propagated using Caco-2 or MA104 cell lines.

The manuscript is written quite comprehensibly, however, there are some issues I would like the authors to respond to.

P2, L49: Instead of citation No. 2 I would recommend the authors to use original article which stated the percentage of intestinal viral infection accountable to the RVA in children (Song and Liu, 2020, is only citing other articles).

P2, L49: “Group A rotavirus (RVAs) are…” please correct as “Group A rotaviruses (RVAs) are…”.

P2, L50: The citation No. 3 (Yang, Wang, Wang et al., 2007) is not appropriate. Please, use original article stating the prevalence of RVA worldwide (e.g. Parashar et al., 2003 or 2006).

P3, L116: The sequence of VP6-R primer is not correct. Please, correct according to the cited article my Matthijnssens et al., where the correct sequence is 5´-GGTCACATCCTCTCACT-3´. It is not possible to reverse the sequence of the primer, as it wouldn´t work.

P4, L164-165: “A second PCR run was performed … using a cocktail of primers.” According to the methods described here, the second PCR is done only with the genotype specific primers (AT8, BT1, CT2, DT4, ET3 and FT9W for G-genotyping; 1T-5T for P-genotyping). In the 2nd PCR run, however, the End9 and con3 have to be used for G-typing and P-typing, respectively.

Please, add this information in the text as well as in the respective Tables 2 and 3.

P5, L185-191: The authors state that the PCR products were sequenced and their sequences compared to those strains available in the GenBank.

Why there is no information about the result of this analyses. Are the 34 RVA strains detected during the 17 months of rotavirus surveillance identical? If not, the phylogenetic analysis might be interesting. And if the detected isolates are similar to a high extend, I would recommend the authors to add the whole genomic sequence of the one described RVA strain 144.

P5, L194: “… accession numbers for the novel sequences …” The sequences described in this manuscript are not novel, as their similarity to other known isolates from the GenBank reached 99.9%. Next, the authors did not describe, which primers were used for the preparation of the whole sequences of VP7, VP4 and VP6 segments. I assume that the VP7 segment was prepared with Beg9/End9 primer pair, but as for the other two segments, the information is missing.

P7, L241: “Caco-2 and MA104 cells were injected with a subset…”. I guess there is a typing mistake – the cells were most probably “infected”, not “injected”. Please, correct.

P8, L249: “… developed in Caco-2 cells on the first and second passages.” What was developed? I presume that the correct sentence would be “… developed CPE in Caco-2 …”.

Another question – Were the strains (n=5) propagated successfully on Caco-2 cells the same as the strains propagated on MA104 cells or different?

P8, L255-257: “… G1 was the most typically recognized G genotype …”, “… G1P[8] was found to be the single most common genotype …”. As the G1P[8] was the only detected genotype, I do not thing that expressions as “typically recognized” or “most common” are suitable.

P12, L293-295: In caption for Figure 6, the explanation of the red arrow use should be added.

P12, L298-299: “The current study is one of the handful research projects on rotavirus disease in children under the age of five that has been conducted in Egypt [11, 23-26].” The meaning of this sentence is a little bit upset. It looks like there were many research projects on rotavirus conducted in Egypt, while the cited research was done in Mozambique or India. Please, either change the citations and leave only Egyptian research projects, or rephrase the whole sentence.

P13, L325: “… G1P[8] was reported to be more dominant than G3P[8] in most reports [11, 26] …” The discussion should include and possibly try to explain the fact, that in 2015-2016 the prevalence of G3P[8] in Cairo was 27%, G1P[4] 18.9%, G1P[6] 13.5% and G9P[8] 10.8%. The genotype G1P[8] was detected in 29.7% which is a great difference when compared with 100% detected in 2018-2020.

P14, L412: The citation No. 11 is not “In press” anymore. It was already published. Please, correct.

Reviewer 2 Report

Review:

Isolation, Propagation and Molecular Characterization of 2 Human Rotavirus Prevailing in hospitalized Children with 3 Gastroenteritis in Egypt

The authors describe a two-year surveillance study of children in two hopitals. RV was found in 14.8 % (n=230) samples and were all identified as G1P8 strains. The authors did a phylogenetic study on VP7, VP4 and VP6 but its not clearly discussed. The VP6 genotype is also not identified. Two cell lines were also used for isolation of rotavirus.

Comments:

Title:  I would say genotyping instead of molecular characterisation

Simple summary:

Line 18: top leading is repetitive. Remove one.

Line 18 – 19: please rephrase “hitting the Egyptian children”

Introduction:

Line 53: Please update available vaccines.

Line 60: correct: greater strain variety

Materials and methods:

Line 114: remove optimal

Table 1 can be written as text

Line 130: the authors do not list which samples were used to infect the mammalian cells. Did they thy all the positive samples? I assume the sample sample set was used to infect MA104 cells than the Caco-2 cells for comparison?

Line 185: were all PCR products sequenced? Why is there only 1 accession number per segment?

Line 198: A NJ tree is not sufficient for publication. The trees need to ML and commonly the rotavirus name is included in the trees.

Results

Line 216: for rotavirus

Line 222-5: Please clarify

Line 241: infected

Line 244: please give more information about which strains were rescued on which cell line

Line 268: please describe which strains were related to the study strains

Figure 5: the images are out of focus

Trees: please supply better quality images

Discussion

Line 301: “tracked from birth to 5 years” Please clarify as the authors stated that samples were taken over two years

Line 327: Please clarify: “existence of a single genotype which was the most often genotype”

Reviewer 3 Report

This manuscript describes molecular surveillance of human RVAs in hospitalized children in Egypt. However, this manuscript has little novelty and scientific soundness, because study design is poor. This manuscript will need to resubmit by including further mass data.

Major problem

It is too small of numbers, collection periods, and places of samples to survey RVA infections in children in Egypt. You should reconsider study design corresponding to your objective.

You should change Table 1-4 into supplementary Table 1-4.

You should remove Figures 2 and 3, because the data represented in Table 5.

Figure 5 is not clear. You should show clearer photos.

Figure 6

People cannot understand the meanings of this figure.

You should show strain name, collection country, collection year, and G and P genotypes, not GenBank accession numbers in this figure.

Reviewer 4 Report

The authors in this report aimed to isolate the most common strain of rotavirus infection prevailing in Egyptian children and to characterize it. By the use of Caco-2 and MA104 cell lines, and fecal samples obtained, authors determined the most common strain found in the fecal samples was Rotavirus G1P[8]. The sample size is fine and results clearly reported. The study is interesting however, too objective.   

1. Is this a cohort study?  If it is, please mention it clearly in the introduction section.       

2. A little background about Caco-2 and MA104 cell lines should be provided in the results section.                                                                                                                      

Round 2

Reviewer 1 Report

The authors improved the manuscript greatly but there are still minor issues demanding explanation.

P7, L262-264: From the text I understand that five RVA strains showed CPE on MA104 cell line (three isolates in the second passage and two in the third passage). The same five RVA strains showed CPE in Caco-2 cells, only the CPE was visible as soon as in the first passage and lasted to the third passage. However, the number of RVA isolates successfully propagated on the Caco-2 cells was the same as on MA104 cells (n=5). Then the sentence “Caco-2 cultures supported rotavirus proliferation of a substantially higher number of isolates than MA104 cells, but the difference was not statistically significant (Figure 2)” should be corrected. The same applies to the Discussion (P13, L351-353).

P8, Table 6: The numbers of RVA strains successfully propagated in Caco-2 cells contain RVA206 twice and do not contain RVA163. Please, correct.

Discussion P13, L355-356: “This remarkable property of rotaviruses makes them promising 355 tools for being used as oncolytic agents’ time ahead.” I do not understand which property of rotaviruses the authors are talking about. Please, explain.

Reviewer 2 Report

Please see my comments below:

In my opinion, the phylogenetic analyses contribute very little to the paper since only one sample out of 34 was sequenced, seemingly after rescue from mammalian cells. That leaves prevalence and genotyping as well as the attempts to rescue. The authors didn't discuss what they intended for the rescued viruses. However, if they wanted to describe the genetic diversity (or lack thereof) in the area it would have been more suitable to sequence the 34 sequences before rescue and then properly discuss the results.

Dear Professor,

Thank you for valuable comments.

All comments were considered with a great interest. Thank you so much

The comments and our response:

1.     I would say genotyping instead of molecular characterization

Authors: We believe your suggestion will be more objective, so the title has been changed.  

2.     Line 18: top leading is repetitive. Remove one.

Authors: the repetitive was removed, (Page 1 Line 18).

3.     Line 18 – 19: please rephrase “hitting the Egyptian children”

Authors: The sentence was changed into “beating the Egyptian children”, (Page 1 Line 18).

2nd round: Please change beating to affecting

4.     Line 53: Please update available vaccines.

Authors: Line 53 mentions global rotavirus vaccines (Rotarix® and RotaTeq®), which are available in Egypt, but the others are local vaccines. In general, we updated the sentence to include other local vaccines, (Page 2 Lines 58-65).

5.     Line 60: correct: greater strain variety

Authors: It was corrected, (Page 2 Line 68).

6.     Line 114: remove optimal

Authors: It was removed, (Page 3 Line 120).

7.     Table 1 can be written as text

Authors: Your suggestion is OK, but we expect that the mentioned table, rather than the text, may be easier for the reader.

8.     Line 130: the authors do not list which samples were used to infect the mammalian cells. Did they thy all the positive samples? I assume the sample sample set was used to infect MA104 cells than the Caco-2 cells for comparison?

Authors: All PCR-positive samples were used to infect mammalian cells, and this was changed in the text for greater clarity, (Page 3 Line 136).

9.     Line 185: were all PCR products sequenced? Why is there only 1 accession number per segment?

Authors: Not all PCR products were sequenced; the accession numbers listed here are only for a single rotavirus strain, RVA144, which was isolated using two cell lines and has a high CPE. The manuscript's paragraph was changed to improve clarity (Page 5 Line 191).

2nd round: Why was only 1 out of 34 strains sequenced and how was this one chosen? Was sequencing done from the sample or the viral infection in the mammalian cells? There is a good chance some changes occurred during rescue and it would be important to state the source. The phylogenetic analyses would be much more relevant if it was based on all the positive samples.

10. Line 198: A NJ tree is not sufficient for publication. The trees need to ML and commonly the rotavirus name is included in the trees.

Authors: the tree was modified with rotavirus names (Page 6 Line 212).

2nd round: There are no bootstrap values available for the trees so it's impossible to assess the statistical relevance of the clustering. Also, please state how many iterations were done for the bootstrap analyses. It should be at least 1000.  The phylogenetic analyses are never discussed in the paper.

11. Line 216: for rotavirus

Authors: It was corrected (Page 6 Line 229).

12. Line 222-5: Please clarify

Authors: It was clarified (Page 6 Lines 235-238).

13. Line 241: infected

Authors: It was corrected (Page 7 Line 252).

14. Line 244: please give more information about which strains were rescued on which cell line

Authors: The strains that were rescued on each cell line were added to Table 6 (Page 8 Line 284).

15. Line 268: please describe which strains were related to the study strains

Authors: It was described (Page 8 Line 281).

2nd round: Please name the closest relatives to the study strains.  Were they from the same region? In the VP4 tree it looks like the study sequence grouped with strains from the US which were also from tissue culture. This makes it hard to assess the genetic relatedness as stated in line 274.

16. Figure 5: the images are out of focus

Authors: It was improved as much as possible, (Page 8 Line 288).

17. Trees: please supply better quality images

Authors: It was improved as much as possible, (Pages 9,10,11).

18. Line 301: “tracked from birth to 5 years” Please clarify as the authors stated that samples were taken over two years

Authors: The sentence was changed to make it clearer: "a cohort of 230 Egyptian children ranging in age from birth to five years" (Page 12 Line 302).

19. Line 327: Please clarify: “existence of a single genotype which was the most often genotype”

Authors: The sentence was changed to make it clearer (Page 12 Lines 327,328).

Reviewer 3 Report

The revised manuscript has not been improved corresponding to reviewer's comments.

It is too small of numbers, collection periods, and places of samples to survey RVA infections in children in Egypt.

Therefore, these data is not reflected the status of RVA infection in Egypt.

You should change Table 1-4 into supplementary Table 1-4, because authors is used data from other researchers, not your original data.

Figure 2 is not clear. Particularly, the background of photos is a serious problem.

Figure 3 

The authors should discuss about characters of the isolated RVA using these data.